A comparative analysis of variants of machine learning and time series models in predicting women’s participation in the labor force

Elstohy Rasha 1 2 rashastohy@nctu.edu.eg
Aneis Nevein 3
http://orcid.org/0000-0002-3729-2778 Mounir Ali Eman 4
1 Department of Information Systems, Obour Institutes , Al-Sharqia, Al-Sharqia , Egypt
2 Department of Information communication Technology, New Cairo Technological University , Cairo, New Cairo , Egypt
3 Basic Sciences Department, Obour Institutes , Al-Sharqia, Al-Sharqia , Egypt
4 Scientific Computing Department, Faculty of Computers and Artificial Intelligence, Benha University , Al-Qalyubia, Al-Qalyubia , Egypt
Moparthi Nageswara Rao
Electronic publication date: 2024 Nov 11
Publication date: 2024
Volume: 10
Electronic Location ID: e2430
Received 2024 May 1; Accepted 2024 Sep 26
Copyright: © 2024 Elstohy et al.
Copyright year: 2024
Copyright holder: Elstohy et al.
License: This is an open access article distributed under the terms of the Creative Commons Attribution License, which permits unrestricted use, distribution, reproduction and adaptation in any medium and for any purpose provided that it is properly attributed. For attribution, the original author(s), title, publication source (PeerJ Computer Science) and either DOI or URL of the article must be cited.
License URL: https://creativecommons.org/licenses/by/4.0/

Keywords: Machine learning, Time series, Women employment, Crisis times, Labor force, Employment rate, Forecasting, Analysis

Funding: The authors received no funding for this work.

==============================
Labor force participation of Egyptian women has been a chronic economic problem in Egypt. Despite the improvement in the human capital front, whether on the education or health indicators, female labor force participation remains persistently low. This study proposes a hybrid machine-learning model that integrates principal component analysis (PCA) for feature extraction with various machine learning and time-series models to predict women’s employment in times of crisis. Various machine learning (ML) algorithms, such as support vector machine (SVM), neural network, K-nearest neighbor (KNN), linear regression, random forest, and AdaBoost, in addition to popular time series algorithms, including autoregressive integrated moving average (ARIMA) and vector autoregressive (VAR) models, have been applied to an actual dataset from the public sector. The manpower dataset considered gender from different regions, ages, and educational levels. The dataset was then trained, tested, and evaluated. For performance validation, forecasting accuracy metrics were constructed using mean squared error (MSE), root mean squared error (RMSE), mean absolute error (MAE), mean absolute percent error (MAPE), R-squared (R2), and cross-validated root mean squared error (CVRMSE). Another Dickey-Fuller test was performed to evaluate and compare the accuracy of the applied models, and the results showed that AdaBoost outperforms the other methods by an accuracy of 100%. Compared to alternative works, our findings demonstrate a comprehensive comparative analysis for predicting women’s participation in different regions during an economic crisis.

Introduction

Women’s participation in the labor force is a key factor in economic growth and sustainable development (Ministry of Labour and Employment, 2022). Therefore, participation rates indicate the potential for a country to grow more rapidly in recent years (Hassan, Khalifa & Shoieb, 2022). As of March 2020, the economy has been significantly impacted by crises such as pandemics and wars. Forecasts indicated that the unemployment rate would peak at 3.98% in the second quarter of 2020. Even the most pessimistic projections suggested that the unemployment rate for the year 2020 would not exceed 7%, but despite this, the employment rate still declined (El Assiouty & Assiouty, 2022).

As a result, the extensive literature on employment forecasting has explored numerous statistical and economic methods. However, there has been a recent shift towards embracing machine learning models, which offer the advantage of being free from limitations associated with traditional approaches. Forecasters have increasingly turned their attention to machine learning (ML) models for employment forecasting, harnessing the benefits they provide. Hence, recent studies indicate that artificial intelligence and machine learning algorithms, in particular, demonstrate exceptional performance when dealing with extensive datasets (Trabay, Gharibi & Abd-Elhafiez, 2023; Kesornsit & Sirisathitkul, 2022).

Machine learning is an algorithmic approach employed to build empirical models based on datasets, falls under the category of data-driven modeling (Celbiş, 2023). These algorithms are customized for extracting critical information from extensive recorded data, leading to enhanced performance and accuracy of predictions (Sakr et al., 2018; Nkongolo, 2023). Other articles demonstrate the popularity of statistical models, particularly autoregressive integrated moving average (ARIMA) and vector autoregressive (VAR) models, as widely adopted techniques in the field of time series forecasting (Ahamed et al., 2022; Nkongolo, 2023)

Only a limited number of studies have explored the combination of ML and statistical time-series models to predict employment rates concerning gender or education. In this regard, this study attempts to employ multiple machine learning models to forecast the direction of Egyptian female and male employment over the next two years based on data collected from an official governmental agency before and after the pandemic. This study examines labor market transitions for women in Egypt during crisis times using machine learning models. The models utilized include AdaBoost (Viljanen & Pahikkala, 2020), random forest (Ahamed et al., 2022) linear regression (Alsharef et al., 2022), K-nearest neighbor (KNN) (Arjaria et al., 2022), neural network (Gogas, Papadimitriou & Sofianos, 2022), and support vector machine (SVM) (Alsubai, 2023) in addition to ARIMA (Vollmer et al., 2021) and VAR (Davidescu, Apostu & Paul, 2021) as time series models.

To ensure accurate predictions, data were preprocessed, relevant features were extracted, and the most correlated features were selected. Subsequently, prediction experiments were conducted using various models to identify the most accurate ones. To achieve this, model validation using a Dickey-Fuller test was conducted during the testing phase. The article is organized as follows: a literature review is briefly presented, “Materials and Methods” explains the proposed model, data gathering, data analysis, preprocessing, and feature extraction, followed by “Results”, “Discussion” describes analysis results and performance comparison among various models, and “Conclusions” concludes the article.

Many researchers have presented various techniques for predicting employment and unemployment rates. Cook & Smalter Hall (2017) developed four distinct neural network architectures to predict the U.S. unemployment rate. Their findings surpassed the benchmark set by the Survey of Professional Forecasters (SPF).

Similarly, Kreiner & Duca (2020) utilized artificial neural networks (ANN) to forecast the U.S. unemployment rate. The models were compared with ARMA, STAR, and three different ANNs. The results demonstrated that radial basis function neural networks (RBFNN) statistically outperformed all other models.

Stasinakis et al. (2016) employed RBFNN, Kalman filters, and support vector regression to forecast the U.S. unemployment rate. Their suggested models were compared against an ARMA, a STAR, and three different ANNs. Results demonstrated that the RBFNN statistically outperformed all the other models. In the UK, Floros (2005) utilized ARIMA and Generalized AutoRegressive Conditional Heteroskedasticity (GARCH) models developed to predict the unemployment rate. In Italy, Naccarato et al. (2018) employed ARIMA and VAR models with official data and Google Trends query rates, results showed a lower forecast error for VAR model.

Within the labor market of Hong Kong’s construction sector, Wong, Chan & Chiang (2005), Lai et al. (2014) utilized ARIMA techniques to examine and predict vital metrics, such as employment quantity, productivity, rates of unemployment, underemployment, and actual wages. In the European Union (Celbiş, 2023), methods such as Box‒Jenkins and Time Series Regression with ARIMA Noise, Missing Observations (TRAMO)/Signal Extraction in ARIMA Time Series (SEATS) have been employed to predict the unemployment rate. In specific European countries, Germany’s unemployment rate was predicted using the Box–Jenkins methodology with ARIMA and VAR models (Jelena, Ivana & Zorana, 2017), Greece utilized seasonal autoregressive integrated moving average models (SARIMA) models for dynamic and static processes (Dritsaki, 2016), and Slovakia used ARIMA and GARCH models (Rublíková & Lubyová, 2013)

A statistical sample of unemployment data obtained from the Chinese National Bureau of Statistics (Liu & Li, 2022) was used as the basis for this study. A labor unemployment prediction model was developed by combining both neural networks and time-series models. This combined approach is used to forecast the labor unemployment situation in China from 2022 to 2030. The results demonstrate that the fitting impact of the combined model outperforms that of the individual models, highlighting the superiority of the combined model. Table 1 provides complete overview of the literature. Although there is an extensive body of literature forecasting unemployment rates in various nations, few studies have focused on predicting employment rates in developing countries. Egypt, in particular, has received limited attention in such studies, with most research focusing on the national level or specific cases. In this study, we investigated employment rates for women compared to men over the past 5 years. We aimed to develop hyper prediction models and assess the accuracy of selected algorithms in this specific context.

Table 1 Summary of related works.

Study	Country/REGION	Models used	Key findings	
Cook & Smalter Hall (2017)	United States	Neural Networks	Neural networks superior for unemployment rate prediction	
Kreiner & Duca (2020)	United States	ANN, ARMA, STAR	RBFNN superior for unemployment rate prediction	
Stasinakis et al. (2016)	United States	RBFNN, Kalman filters, SVR, ARMA, STAR, ANN	RBFNN superior for unemployment rate prediction	
Floros (2005)	United Kingdom	ARIMA, GARCH	Early application of ARIMA and GARCH for unemployment prediction	
Naccarato et al. (2018)	Italy	ARIMA, VAR	Impact of Google Trends data on unemployment forecasting	
Lai et al. (2014)	Hong Kong	ARIMA	Application of ARIMA for various labor market metrics	
Celbiş (2023)	European Union	Box-Jenkins, TRAMO/SEATS	Overview of methods used in EU for unemployment prediction	
Jelena, Ivana & Zorana (2017)	Germany	ARIMA, VAR	Application of ARIMA and VAR for German unemployment	
Dritsaki (2016)	Greece	SARIMA	Application of SARIMA for dynamic and static unemployment patterns	
Rublíková & Lubyová (2013)	Slovakia	ARIMA, GARCH	Application of ARIMA and GARCH for Slovakian unemployment	

Materials and Methods

This section provides a detailed overview of our research methodology. The architecture, shown in Fig. 1, consists of the workflow of the procedure. The decomposition phase begins with the main decomposition characteristics, followed by the feature selection phase, the proposed models, and finally the forecasting results.

Figure 1 Framework architecture.

Material and dataset

CAPMAS reports (2018–2022) were released as a quarterly report on the labor force, encompassing information about workforce volume, labor participation, and the assessment of employment and unemployment categorized by gender and geographical distribution. This research focuses on the labor force survey bulletin of CAPMAS from the fourth quarter (October–November–December) of 2018 to the second quarter of 2022 (CAPMAS, 2018, 2019, 2020, 2021, 2022).

The dataset consists of 1,731 records and 14 attributes. The data chosen were statistically analyzed, preprocessed, and then features were selected before applying predictive models. In the study, the term “manpower or employment” encompasses the entire population, excluding individuals aged less than 6 years, individuals aged 65 years or older who are not engaged in work, unwilling to work, and not actively seeking employment, and individuals who are completely disabled. These three categories are classified as part of an economically inactive population. The dataset used in this study includes the following attributes: 1) Characteristics of the Egyptian civilian labor force.

2) Extent of employment across different geographical regions of the country.

3) The distribution of employed individuals in different areas was based on factors such as gender, age, educational background, employment status, occupation, economic activity, sector, job stability, and working hours. A screenshot of our dataset is shown in Fig. 2.

Figure 2 Dataset sample.

Statistical analysis

For a thorough analysis of the captured dataset, statistical analysis was conducted to better understand the relationships between historical data and its attributes. This analysis illustrates the employment rates for both males and females in various regions, educational levels, and age groups, as shown in Figs. 3A, 3B.

Figure 3 Employment rate for male, female according to region and age.

Data preprocessing

To avoid missing data and group each subset for data preparation in the subsequent phases, the data were preprocessed. Table 2 illustrates the assessment of the data quality properties for labor force parameters for different regions, education levels, education distribution across regions, age-related workforce distribution by region, and education level. The assessment utilized statistical methods, such as mean, mode, median, dispersion, minimum, and maximum, for data cleaning and correction.

Table 2 Data quality assessment properties.

Distribution name	Mean	Mode	Median	Dispersion	Min.	Max.	Missing	
Labor force contribution percentage in all regions	0.16	0.15	0.16	0.10	0.14	0.20	0 (0%)	
Labor force actual contribution (million) in all regions	4.24	4.10	4.24	0.10	3.30	5.17	0 (0%)	
Labor force percentage according to region	0.15	0.09	0.16	0.33	0.08	0.20	0 (0%)	
Labor force actual contribution (million) in specific region	0.63	0.31	0.73	0.37	0.31	1.01	0 (0%)	
Education percentage	0.09		0.09	0.10	0.08	0.11	7 (44%)	
Total contribution for each region (million)	0.07		0.08	0.18	0.05	0.09	7 (44%)	
Total contribution according to age range (percentage) for each region	0.02	0.01	0.01	0.55	0.01	0.04	0 (0%)	
Total contribution for each age according to region (million)	0.01	0.00	0.01	0.74	0.00	0.03	0 (0%)	
Employment total no (million)	26.63	26.62	26.62	0.03	24.12	27.83	0 (0%)	
Date	43,882.06	43,922.00	43,922.00	~4 years	43,101.00	44,562.00	0 (0%)	
Region		North Urban		0.685			0 (0%)	
Educational level		Cannot read/write		0			7 (44%)	
Age		15–19		0.685			0 (0%)	

Feature extraction

Feature extraction involves the selection and transformation of the most crucial features from the data to be utilized in developing predictive models, whether using statistical methods or machine learning models (Kumar & Rajini, 2019). For the employment dataset, only specific columns were used for the comparative analysis of the model. These columns include the date by quarter, actual labor force (in millions) in all regions, and total contribution for each age group by region (in millions). Principal component analysis (PCA) was employed to extract the relevant features from the employment dataset (Ahamed et al., 2022). This approach effectively reduces the computational complexity, enabling the creation of generalizable models and optimization of the storage space. By addressing redundancy, the PCA selects a subset of relevant patterns from the original dataset based on their significance. Additionally, it aids in identifying attributes with the highest significance for recognition and subsequent inclusion in predictive analysis. Table 3 presents the correlation matrix, which summarizes the attributes used in the dataset.

Table 3 Pearson correlation for dataset feature extraction.

n	Correlation	Feature 1	Feature 2	
1	0.992	Employment total no (million)	Date	
2	−0.824	Employment total no (million)	Labor force contribution percentage in all regions	
3	−0.795	Total contribution for each region (million)	Employment total no (million)	
4	−0.74	Employment total no (million)	Labor force actual contribution (million) in all regions	
5	−0.527	Employment total no (million)	Labor force actual contribution (million) in specific region	
6	−0.51	Total contribution for each age according to region (million)	Employment total no (million)	
7	−0.312	Employment total no (million)	Labor force percentage according to region	
8	−0.304	Employment total no (million)	Total contribution according to age range (percentage) for each region	
9	−0.301	Education percentage	Employment total no (million)	

Proposed model validation technique

Following the data preprocessing and feature selection phases, various machine learning predictors were used to predict the next 12 months, the same dataset was used using time series models, with a focus on accuracy assessment using MATLAB. These models were fed a data frame as the input, enabling direct inference of the data. where the dataset was split into 80:20 training and testing sets for the prediction phase, this procedure ensures that all algorithms undergo a fair and equitable evaluation. The same dataset is employed for all algorithms, and forecast errors remain completely blind to the held-out data, mimicking a real forecasting scenario.

Results

The results and outcomes prediction of the women participation in labor force are presented in this section

Comparative analysis

More about the empirical analysis, Fig. 3 shows that female participation rates are generally lower than those of males in the last recorded quarter of 2022. Generally, when reviewing the employment rates according to age groups, it is evident that the employment rates for the age groups 30–39 and 40–49 are the highest among males, reaching 92.8% and 93.1%, respectively. On the other hand, the employment rates for the age groups (40–49) and (50–59) are the highest among females, reaching 19.9%. By region, the lowest percentage of females participating in the labor force is found in the countryside of Upper Egypt, amounting to 10.5%. From an educational perspective, the data showed that the rate was higher than the average for individuals holding university qualifications and above. The study revealed a contribution level of up to 44.6%, followed by qualifications ranging from above intermediate to below university, with an average of 34.6%.

Forecasting performance comparison

To accurately predict future employment rate values and analyze trends, we used different performance metrics as mentioned before, to assess model accuracy; in this regard, it is crucial to utilize the entire collected dataset instead of relying on samples when employing forecasting performance methods. The metrics used to evaluate forecasting accuracy include mean squared error (MSE) (Davidescu, Apostu & Paul, 2021), root mean squared error (RMSE) (Ahamed et al., 2022), mean absolute error (MAE) (Kesornsit & Sirisathitkul, 2022), mean absolute percent error (MAPE) (Davidescu, Apostu & Paul, 2021), R-squared (R2) (Shen, 2022), and cross-validated root mean squared error (CVRMSE) (Sakr et al., 2018), as formulated in Eqs. (1)–(6).

(1) MSE=Σ(yi−pi)2/n

(2) RMSE=√1nn∑i=1(yi−yi^)2

(3) MAE=(1/n)Σ(i=1ton)|yi−y^i|

(4) MAPE=(Σ(IA−FI)/Ax100)/N

(5) R2=1−(RSS/TSS)

(6) CV(RMSE)=RMSE/Y¯

The superior forecast performance of the model is evident in its lower error statistics. Table 4 presents experimental models for comparing their performance in predicting male and female employment rates. After studying the results obtained for females and males, we observe the following: The AdaBoost model performs well for both genders, exhibiting the lowest MSE, RMSE, and MAE values, as well as high R2 values.

Random forest also performs relatively well, with lower MSE and RMSE values but higher MAE values than AdaBoost. KNN and linear regression models perform similarly, exhibiting higher MSE and RMSE values, as well as higher MAE values.

The neural network and SVM models performed poorly, showing higher MSE, RMSE, and MAE values, as well as negative R2 values for the females. In contrast, all models perform better for males compared to females, as illustrated in Figs. 4–6. When selecting region, age, and educational background, it is important to consider that descriptive data is more precise and freer of missing values for male participants compared to female participants. In time, our dataset is considered stationary, ARIMA and VAR time series algorithms were tested and evaluated using the same forecasting accuracy indicators for both male and female genders, as shown in Table 5, followed by Fig. 7. The results in Table 5 summarize ARIMA vs. VAR for prediction accuracy as follows:

The VAR model has a lower MSE of 0.019 for females, suggesting higher accuracy. Similarly, for the male model, the VAR model has a lower MSE (0.278) than that of the ARIMA model (0.479).

According to the RMSE, the female VAR model exhibited a lower RMSE of 0.018 than the ARIMA model’s RMSE of 0.026, suggesting higher accuracy. Similarly, for males, the VAR model has a lower RMSE (0.015) than the ARIMA model’s RMSE of 0.025.

Observing the MAE rate: both the VAR and ARIMA models have the same MAE for females, with values of 0.013 and 0.019, respectively. For males, both the VAR and ARIMA models have the same MAE (0.278 and 0.479, respectively).

Table 4 ML forecasting performance comparison for gender criteria.

Female	
Model	MSE	RMSE	MAE	R2	CVRMSE	
AdaBoost	0.003	0.052	0.017	0.916	8.041	
Random forest	0.006	0.075	0.046	0.83	11.485	
Linear regression	0.009	0.095	0.068	0.725	14.582	
KNN	0.009	0.093	0.065	0.735	14.324	
Neural network	0.037	0.193	0.155	−0.138	29.68	
SVM	0.129	0.359	0.213	−2.927	55.14	
Male	
AdaBoost	0.076	0.275	0.091	0.947	1.456	
Random forest	0.094	0.306	0.193	0.934	1.621	
KNN	0.174	0.417	0.297	0.878	2.209	
Linear regression	0.269	0.519	0.401	0.811	2.75	
Neural network	0.274	0.523	0.402	0.808	2.772	
SVM	0.332	0.576	0.433	0.768	3.052	

Figure 4 MSE for geographical regions.

Figure 5 MSE for age feature.

Figure 6 MSE for education feature.

Table 5 Arima and VAR employment prediction performance measurement.

Female	
Model	MSE	RMSE	MAE	
ARIMA	0.019	0.026	0.019	
VAR	0.013	0.018	0.013	
Male	
ARIMA	0.479	0.025	0.479	
VAR	0.278	0.015	0. 0.278	

Figure 7 VAR vs. ARIMA prediction among regions.

Discussion

In Egypt, despite achieving higher levels of education, women’s participation in the labor force has remained low in recent years. Despite the substantial rise in women’s educational achievements, their involvement in the labor market continues to be relatively limited. Nevertheless, the available data and findings from related studies concerning women’s behavior in the labor market during times of crisis are limited and do not seem to reveal consistent patterns. Moreover, the majority of other nations research articles demonstrate the popularity of statistical models, particularly ARIMA and VAR models, which are widely adopted techniques in the field of time series forecasting. This research develops hyper prediction models and assesses the accuracy of selected algorithms, Meanwhile, forecasting accuracy provides useful information about the goodness of fit of the forecasting model and demonstrates the model’s ability to anticipate future employment rates. The metrics used to evaluate forecasting accuracy include the MSE, RMSE, MAE, MAPE, R2, and CVRMSE. Lower MSE values indicate better model performance. Overall, AdaBoost appeared to be the best-performing model for both the female and male datasets, based on the provided metrics, followed by KNN. It demonstrates lower errors (MSE, RMSE and MAE) and higher predictive power (R2) than other models. However, it is important to note that the selection of the optimal model also relies on the specific requirements of the problem and context in which it is applied. In a Comparing the forecasting accuracy of the ARIMA and VAR time series models, the VAR model outperforms the ARIMA model for both female and male datasets. It consistently demonstrated lower values across all metrics, indicating higher prediction accuracy. To add value and assess potential differences in forecast accuracy between machine learning and time series models, the Dickey-Fuller test was employed (Davidescu, Apostu & Paul, 2021). This test assumes a null hypothesis of no difference and an alternative hypothesis suggesting the presence of a significant difference in forecast accuracy between the two models.

Dickey-Fuller test

As mentioned in previous sections, MSE, RMSE, and MAE represent the error or the difference between the forecasted values and the actual values. Lower values of these metrics indicate a better forecast model, while higher values indicate poor performance. To convert these metrics into accuracy percentages, a threshold or a target value for the forecasted variable should be defined. Then, the accuracy percentage by comparing the forecasted values to the target value as in Eq. (7) should be calculated. Table 6 shows the accuracy comparison after calculating the Dickey-Fuller equation for females based on selected features. Figure 8 summarizes these comparisons.

(7) Accuracypercentage=((Targetvalue−Error)/Targetvalue)∗100

Table 6 Dickey-Fuller performance testing.

Female	
	SVM	Linear regression	AdaBoost	Neural network	Random forest	KNN	
Accuracy_Geographical_region	81.37	91.05	100.00	95.98	93.59	92.79	
Accuracy_ages	94.78	89.06	99.69	34.84	88.14	95.46	
Accuracy_Education	73.37	94.61	82.30	95.04	95.14	73.05	

Figure 8 Machine learning model accuracy %.

From Fig. 8, it seems that AdaBoost is still the best predictive model with 100% accuracy in geographical feature, followed by random forest, which appears to have better performance in relation to educational levels.

Conclusions

Machine learning techniques have demonstrated strong predictive capabilities in various application domains, including forecasting future employment for both men and women. In this article, several machine learning models have been employed to forecast the direction of Egyptian female employment compared to male employment for the next 2 years. The forecasts are based on demographic data collected from the official governmental agency (CAPMAS) before the COVID-19 pandemic in 2018 and after the pandemic in 2022. The collected data is statistically analyzed, preprocessed, and then features are extracted using a PCA model. Comparative analysis concluded that the employment rates for the age groups (40–49) and (50–59) are the highest among females, reaching 19.9%. They contribute up to 44.6% for university certificate holders. However, the lowest percentage of females participating in the labor force is found in the countryside of Upper Egypt, amounting to 10.5%. This disparity in male and female participation rates across different regions and quarters should be a priority for policymakers and stakeholders to ensure inclusive and equitable economic development. Subsequently, we conducted prediction experiments using various machine learning models to determine the accuracy of the best predictive model. Machine learning models such as AdaBoost, random forest, linear regression, KNN, and neural networks have been utilized. Performance evaluation showed that the AdaBoost model performed well for both genders, with the lowest MSE, RMSE, and MAE values and high R2 compared to other ML models. Popular time series statistical models such as ARIMA and VAR were tested for performance, indicating that the VAR model outperforms the ARIMA model for both the female and male datasets, with lower MSE values of 0.013 and 0.278, respectively. Additionally, a Dickey-Fuller test was conducted on the total employment dataset as another performance indicator. The Dickey-Fuller results confirmed that AdaBoost had superior performance with 100% accuracy for geographical features. For future research, it would be beneficial to study various work sectors during the months following the Russian-Ukraine war, considering different demographic characteristics and optimizing performance for models with weak results.

Additional Information and Declarations

Competing Interests

Author Contributions

Data Availability

The authors declare that they have no competing interests.

Rasha Elstohy conceived and designed the experiments, analyzed the data, prepared figures and/or tables, authored or reviewed drafts of the article, write article, select journal, journal template, language check, submit paper, and approved the final draft.

Nevein Aneis collected the data, analyzed the data, prepared figures and/or tables, and approved the final draft.

Eman Mounir Ali performed the experiments, performed the computation work, prepared figures and/or tables, and approved the final draft.

The following information was supplied regarding data availability:

The distribution ratio cumulative data is available at GitHub: https://github.com/dremanmonir/employment/blob/1e6a85c6e68c4f25c86ef9fc69a9af3a765eef36/Distribution%20Ratio%20comulative.xlsx

The code is available at GitHub:

https://github.com/dremanmonir/employment/tree/a5534d985a49193755a06a2daf191820461f183c/models

The data and code are available at Zenodo: dremanmonir. (2024). dremanmonir/employment: first_release (first_release). Zenodo. https://doi.org/10.5281/zenodo.13851512.

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
