# Peer review of "A comparative analysis of variants of machine learning and time series models in predicting women’s participation in the labor force"

_PeerJ Computer Science, doi:10.7717/peerj-cs.2430_

## Round 0.1 · original submission · Major Revisions

Dear Author
Great news for you
There are a few suggestions for the article make it effective and will re-submit once completed suggestions
Thanks
Dr. M Nageswara Rao
AE -PeerJ Computer Science

·

Basic reporting

The language used is mostly clear, but there are a few grammatical and typographical errors that need
correction. For instance, "As of March 2020, the economy has experienced significant effects due to crises such as pandemics and wars" should be corrected for tense consistency. Additionally, phrases like "ducky filler" seem out of context and require clarification or correction.

Some references are cited out of order, and there are inconsistencies in citation formatting that need to be corrected.

Experimental design

The investigation seems rigorous with the application of various machine learning and statistical models. However, more details on ethical considerations regarding the use of data, especially if it involves personal information, should be included.

The methodology is detailed, but there are some areas that need more specificity. For example, the preprocessing steps, the exact parameters used for machine learning models, and the computing infrastructure should be detailed enough to ensure replicability.

The evaluation methods and metrics are adequately described. Metrics such as MSE, RMSE, MAE, MAPE, R², and CVRMSE are used, which are appropriate for the study. However, more explanation on how these metrics were chosen and their significance would be beneficial.

Sources are generally cited adequately, but there are some inconsistencies in formatting, and some references seem to be cited out of sequence. Correcting these issues will improve the readability and credibility of the paper.

Validity of the findings

The impact and novelty of the work are not directly assessed. However, the paper does contribute to the field by exploring the combination of machine learning and statistical models for employment forecasting, which is a novel approach.
 The conclusions are well stated and supported by the results. They address the main findings and suggest future research directions.
 The experiments and evaluations are performed satisfactorily. Various models are tested, and their performances are compared using appropriate metrics. However, the explanation of why certain models performed better should be expanded.
 The argument is well-developed and supported by the data analysis and results. The study meets its goals by demonstrating the effectiveness of machine learning models in employment forecasting.
 The conclusion identifies future research directions but could discuss limitations more explicitly. Addressing potential biases, limitations of the dataset, and model limitations would provide a more balanced view.

Additional comments

The paper is comprehensive and covers a significant amount of ground in employment forecasting. However, some sections need more detail, especially in the methodology and data preprocessing parts. Additionally, there are grammatical and typographical errors that should be corrected for clarity. The flow of the paper can be improved by making smoother transitions between sections. Clarifying the term "ducky filler" or replacing it with the correct term is also necessary.

Reviewer 2 ·

Basic reporting

The paper is well-written and describes the use of models capable of predicting women's employment in times of crisis. While the approach employed is not new, relying on existing algorithms and techniques, the combination of these methods defines a realistic and applicable approach. The results and statistics are well-defined, but a more detailed description of the dataset would enhance the reader's understanding.

Experimental design

The performance validation is well explained, and the provided graphs effectively illustrate  results. This kind of research can serve as a foundational reference for future studies and offers a valuable perspective on how to predict women's employment trends during crises.

Validity of the findings

no comment

·

Basic reporting

1. I suggest adding a detailed summary table of the related studies with appropriate parameters.
2. The overall structure of this paper is good and well explained with proper explanations.
3. State of the art comparison is not sufficient.
4. No comparative analysis more literature survey

Experimental design

NA

Validity of the findings

NA

Additional comments

NA

---

## Round 0.2 · accepted · Accept

Dear Author/Authors

Congratulations we are accepting the article for PeerJ Computer science
please complete the rest of the procedure

Thanks
AE - PeerJ Computer Science

·

Basic reporting

Adequate

Experimental design

No comment

Validity of the findings

Not comment

Additional comments

The author has made the necessary revisions, and the work is now suitable for publication.

Reviewer 2 ·

Basic reporting

The paper shows a professional English language so the understanding of the article is clear. Also the literature is well explained and the concepts are well explained. The introduction is motivating but I feel that more emphasis should be given to the results and also to the general architecture that is well written but probably giving more details will bring advantage to the paper.

Experimental design

The article reflects the title of the paper and even if the general method is described probably is preferable to have a detailed low level description like specifying some functions that have been used or add small pieces of code. The dataset is well described and I don't think that adding more information will be benefit. Also metrics are correctly described but I feel that is missing a general scheme to gives information of what could be the advantages and disadvantages of using each technique. Source are correctly cited.

Validity of the findings

The validity of findings is important because even if the methods are well know and used in literature the paper could give the basis on a more extensive analysis on different context. Results are well stated but it could be also interesting to add information related of what could be the next steps.

Conclusion met the goals set in the introduction but more information of what we are expecting in the future could be useful.

I recommend the authors for this article with some minor modifies because is professionally written and the information captures the attention of the reader..The only weakness that I find is that probably more low-leve information should be provided in order to give more vision of the work that is behind this paper.

·

Basic reporting

all ok

Experimental design

All ok

Validity of the findings

all ok